# Using Task-Evoked Pupillary Response to Predict Clinical Performance during a Simulation Training

**DOI:** 10.3390/healthcare11040455

**Published:** 2023-02-04

**Authors:** Elba Mauriz, Sandra Caloca-Amber, Ana M. Vázquez-Casares

**Affiliations:** 1Department of Nursing and Physiotherapy, Universidad de León, Campus de Vegazana, s/n, 24071 León, Spain; 2Institute of Food Science and Technology (ICTAL), La Serna 58, 24007 León, Spain

**Keywords:** pupil response, mental workload, clinical performance, emergency care, simulation practice

## Abstract

Training in healthcare skills can be affected by trainees’ workload when completing a task. Due to cognitive processing demands being negatively correlated to clinical performance, assessing mental workload through objective measures is crucial. This study aimed to investigate task-evoked changes in pupil size as reliable markers of mental workload and clinical performance. A sample of 49 nursing students participated in a cardiac arrest simulation-based practice. Measurements of cognitive demands (NASA-Task Load Index), physiological parameters (blood pressure, oxygen saturation, and heart rate), and pupil responses (minimum, maximum, and difference diameters) throughout revealed statistically significant differences according to performance scores. The analysis of a multiple regression model produced a statistically significant pattern between pupil diameter differences and heart rate, systolic blood pressure, workload, and performance (R^2^ = 0.280; F (6, 41) = 2.660; *p* < 0.028; d = 2.042). Findings suggest that pupil variations are promising markers to complement physiological metrics for predicting mental workload and clinical performance in medical practice.

## 1. Introduction

Clinical competence, especially in emergency care, relies on demonstrating knowledge, skills, behavior, and judgment in stressful situations [1,2]. Applying these factors in the healthcare environment contributes to high-performance development while improving patient safety and satisfaction, healthcare quality, and clinical outcomes.

Practitioners’ performance in emergency departments may decrease under time pressure or mental effort [3,4,5]. Managing situations with patients in critical conditions may also cause delayed response time, deviations from protocols [6], mental fatigue, and burnout in health providers [3,4,7]. Thus, operators’ training in simulated scenarios is often used as a reliable tool for measuring performance and confirming clinical competencies [8].

Improving a trainee’s performance is strongly associated with diminishing mental workload [9]. To meet the demands of a challenging task, mental resources, or cognitive load, may not be considered a burden that hampers performance [3,4,10]. Since cognitive work overload can mitigate clinical performance under high-pressure situations [4], measuring mental workload plays a pivotal role as an indicator of appropriate performance [3,11]. Therefore, monitoring cognitive load is essential to determine whether a demanding task may exceed the trainee’s capacity resulting in limited learning and improvement.

Traditionally, mental workload measures rely on psychometric rating scales due to their easy administration and sensitivity to determine task demand [12,13]. NASA-TLX and 9-point Paas are the most used scales in simulation settings [14,15]. These instruments provide subjective assessments of the mental effort motivated by the trainee’s thoughts and feelings about a previous task. Although psychometric scales showing a strong correlation with mental effort are valuable tools for measuring work overload, their subjective nature limits their application to simulated clinical environments [16].

Monitoring real-time changes in mental load requires the use of objective metrics such as the psychophysiological response. In particular, complex cognitive tasks drive a response in the sympathetic nervous system that triggers the elevation of vital signs (e.g., blood pressure, heart and respiration rates, temperature, and galvanic skin response) [17,18,19,20,21]. Other variables such as anxiety, muscle fatigue, and pupil response are also associated with the operator’s cognitive effort [3,4,22,23].

The pupillary reaction to the amount of surrounding illumination because of the pupillary light reflex is well-known. However, arousal and changes in mental workload also activate a pupillary response [22]. This response is controlled by the sympathetic pathway of the autonomic nervous system, whereas the parasympathetic pathway regulates the pupil light reflex [23]. When addressing cognitive demands, the control of emotions may increase sympathetic activity to the detriment of the parasympathetic pupil’s light reflex and emotional response [24,25,26,27]. Therefore, noradrenergic arousal can facilitate cognitive control [27,28] by inducing changes in pupil size during the cognitive processing of complex visual tasks [23,29,30,31]. According to the task difficulty, pupil diameter increases in response to the mental workload and constricts as the processing load diminishes [32].

Hence, pupil dilation can be considered a reliable measure to evaluate mental workload and cognitive factors when solving a problem or completing a task. Previous works have demonstrated the potential of pupillary responses for indirect measurement of the cognitive load either in simulation-based or clinical settings [31]. Task-related cognitive workload has been investigated by measuring it in robotic surgery, microsurgery, laparoscopic training tasks as well as trauma resuscitation, catheterization, and medical administration simulation practices [6,10,22,33,34,35,36,37,38].

Since appropriate performance indicators are needed to ensure good clinical practices, monitoring pupillary responses can help to understand cognitive demands and mental workload throughout simulation training. Examining pupil size variations in combination with subjective psychometric measures of cognitive load will contribute to addressing this knowledge gap. Therefore, this work aims to evaluate the correlation between task-evoked pupil dilation, psychometrics, and the trainees’ performance during a cardiac arrest simulation practice.

## 2. Materials and Methods

### 2.1. Participants and Study Design

Second-year undergraduate nursing students participated in the study. There were no exclusion criteria beyond being part of the academic program. During the first four-month period of the course, the students received theoretical and practical classes on basic and advanced CPR maneuvers, learning about shockable rhythms and how to respond to them. Participants practiced simulation training during the course, so they were familiar with high-fidelity simulation mannequins, monitoring devices, and facilities. The sample recruitment process was developed during the teaching period, offering participation in the study during the classes, and providing an online form to access the training. Participation was voluntary and included neither remuneration nor academic compensation. The Ethics Committee of the University of León gave its approval to the study. All participants signed written informed consent, following the indications of the Declaration of Helsinki. Data confidentiality and participant anonymity were also guaranteed.

### 2.2. Data Collection

Participants performed the clinical training in a simulated scenario comprising a cardiorespiratory arrest in which the cardiac rhythm corresponded to pulseless ventricular tachycardia (Resusci Anne QCPR, Laerdal Medical, Stavanger, Norway). Subjects participated individually without receiving any instruction during the study. Nursing students were to apply advanced life support (assessment and stabilization of the victim, identification of the cardiac rhythm, and defibrillation of the patient) according to the recommendations of the European Resuscitation Council.

Before performing the intervention, all participants completed a sociodemographic questionnaire. Physiological parameters—peripheral temperature (HyWell SZHIT003, Guangdong China), heart rate, systolic and diastolic blood pressure (Omron M6 Confort IT, Kyoto, Japan), oxygen saturation (Beijing Choice Electronic Technology Co., Ltd., Beijing, China), and pupil size (Pupil Labs GmbH, Berlin, Germany)—were recorded at the beginning and the end of the simulation practice. After finishing the task, the participants completed the NASA workload questionnaire (NASA Task Load Index (NASA TLX)) [15,39].

During the intervention, one of the investigators completed a 10-item checklist Scoring ranges from 0 (lowest score, most incomplete performance) to 10 (highest score or adequate performance) based on the quality of the observed performance (Appendix A). The checklist version was an adaptation of the Basic Life Support (BLS) adult CPR (Cardiopulmonary Resuscitation) and AED (automated external defibrillator) skills testing checklist from the American Heart Association. The adaptation considered particular features of the proposed scenario and the previous academic formation of participants [40].

### 2.3. Data Organization and Analysis

Pupil diameters were recorded with a sampling rate of 200 Hz using an eye tracker device with two video cameras, for the environment (30Hz@1080p) and for pupillary movements, dilations, and contractions, respectively (200 Hz @ 192 × 192px) (Pupil Labs GmbH, Berlin, Germany). The recording quality was adjusted with a spatial 5-point calibration task. Pupil size was monitored throughout the entire duration of the exercise. The data from the device were exported using Pupil Player v3.1.16 software, and the minimum and maximum diameter values were obtained for each participant. Subsequently, the pupil diameter variation was calculated by subtracting the maximum and minimum diameters (Figure 1).

The multidimensional National Aeronautics and Space Administration Task Load Index (NASA-TLX) assessment instrument provided a global workload score for the given task. The questionnaire contains six scales rating for mental, physical, and temporal demands, for performance, effort, and frustration. Participants self-evaluated their workloads by completing a weighting phase before the simulation task and a scoring phase after the task. The NASA TLX has been previously applied in high-fidelity simulation environments under stressful situations [37]. The internal consistency of NASA TLX has been demonstrated for a sample of 398 Spanish workers (Cronbach alpha coefficient α = 0.69) [41].

Finally, the Visual Analog Scale of stress (VAS) was applied to measure the participants’ self-perceived stress levels before and after the cardiac arrest simulation completion. Each participant indicated their stress level by completing a 100 mm line with equidistant points between 1 (very little) and 10 (very much).

### 2.4. Data Analysis

Continuous variables were expressed as mean values ± standard deviation (SD). The Kolmogorov–Smirnov test was applied to determine data normality. Differences between student groups were analyzed with independent *t*-tests. A comparison between pre-test and post-test simulation scores was completed using Student’s *t*-test paired and a Wilcoxon signed rank test for normal and non-normal distribution variables, respectively. Bivariate correlations were used to assess associations between physiological, pupillary metrics, and workload variables and CPR quality parameters. Testing was conducted on several multiple linear regression models in which differences in pupil variations were considered as dependent variables and the rest of the variables (physiological and workload dimensions) as predictors. The software package SPSS for Windows version v.26 (IBM SPSS, Inc., Chicago, IL, USA) was used for data analysis. A *p*-value of <0.05 was set as representing statistical significance for all analyses.

## 3. Results

Forty-nine nursing students (40 female; mean age = 22.59 ± 2.23 SD years) participated in the study (Table 1). The analysis of sociodemographic data showed no significant differences regarding the age and sex of participants. Only 34.7% of the students had received training in Basic Life Support (BLS) beyond the academic content. Twenty-eight participants out of forty-nine obtained scores superior to 5 on the performance checklist, with the mean score being 5.32 ± 2.084. All the participants had previous experience in simulation-based resuscitation training and were competent in providing BLS.

### 3.1. Psychophysiological Parameters

Physiological parameters such as peripheral temperature, heart rate, and oxygen saturation increased after the simulation training, while systolic and diastolic blood pressure records were slightly inferior. Heart rate values showed statistically significant differences between pre-test and post-test measurements; see Table 2. The comparison of participants regarding the checklist scores revealed that those with higher scores presented lower values in all physiological parameters in both pre-and post-test determinations (Table 3).

The participants obtained similar pre-test (4.5 ± 2.44) and post-test (4.8 ± 2.48) stress levels, although slightly higher scores were observed after completing the simulation procedure. Concerning task performance, participants reported lower stress levels at the end of the task regardless of their scoring on the checklist (Table 3).

Pupil responses ranged from 0.41 mm and 9.27 mm, with a mean pupillary diameter difference of 3.64 ± 1.21 (*p* < 0.000). Differences between maximum and minimum pupil diameters were statistically significant (t = −19.278, *p* < 0.000). The analysis of task performance revealed that participants with higher rates on the checklist showed lower pupil diameters than participants with lower checklist scores (<5). Similarly, the relationship between pupil diameter differences and temporary workload demands (r = −0.289, *p* < 0.047), systolic blood pressure pretest (r = −0.315, *p* < 0.029), and systolic blood pressure post-test was negative and moderate, while a positive and moderate association was observed for oxygen saturation (r = 0.324, *p* = 0.027) and stress levels (r = 0.344, *p* < 0.017).

### 3.2. Task Performance and Workload

NASA-TLX weighted scores ranged from 175 (low workload) to 1380 (high workload), showing a mean value of 803 ± 272 [39]. Frustration was the category most rated by participants (*p* < 0.000), while the physical workload was the lowest rated (*p* < 0.001). Concerning task performance, participants with checklist scores <5 displayed higher workload scores on the global score and all demand categories—mental, physical, temporal, performance, effort, and frustration—although the difference was not statistically significant (Table 4).

### 3.3. Pupil Diameter vs. Workload

The difference in pupil diameter through the simulation procedure showed moderately significant associations with other physiological parameters such as systolic pressure and peripheral temperature. Pupil diameter differences were also negatively associated with temporal workload demands. A multiple regression analysis modeled the relationship between pupil diameter differences and (i) pre-test blood systolic pressure, (ii) heart rate, (iii) workload (physical, mental, and temporal demands), and checklist scores (Table 5). The significant regression equation using pupil diameter as a dependent variable explained 28% of the variance (R^2^ = 0.280; F (6, 41) = 2.660; *p* < 0.028; d = 2.042).

## 4. Discussion

This study investigated the impact of task-evoked pupillary response as an indicator of mental workload, in the performance of an unannounced simulated in-hospital cardiac arrest scenario. Since physiological parameters play a pivotal role in clinical skills acquisition, blood pressure, heart rate, peripheral temperature, and oxygen saturation are considered valuable markers of healthcare simulation training. Thus, this study focused on exploring the reliability of pupil diameter changes and other physiological determinations compared with subjective measurements based on self-evaluated workload questionnaires.

Our results confirm the association between cardiovascular response and task performance during simulation practices [3,4,21]. All physiological metrics, except for blood pressure, increased after the training, where heart rate differences statistically significant between pre-test and post-test measurements. The psychophysiological arousal resulting from the activation of the sympathetic nervous system and the subsequent hormonal response to stress explained the change in cardiac activity [42]. Moreover, the performance in the simulation task was inferior in those participants with higher heart rate measures in the post-simulation stage. These findings support the use of heart rate and heart rate variability as acceptable methods to predict cognitive load when performing a difficult task.

Regarding pupil size measures, pupillometry can take advantage of the physiological characteristics of the human eye to investigate the effect of cognitive workload during a demanding task [4,43]. In this study, pupil diameter changes occurring throughout the simulation procedure revealed a positive and moderate association with the participants’ stress levels scores. The physiological response involving pupil dilation has been previously reported in training exercises as the level of stress increases [29,32,44]. The perception of the simulation setting as stressful is congruent with the cardiac response experienced by participants and the increase in heart rate in the post-simulation stage [45,46]. In this sense, the increase in pupil diameter differences correlated well with lower systolic blood pressure values, thus confirming the effect of pupil responses on the psychophysiological stress experienced by participants during the training. Therefore, the contribution of pupil size as a complement to psychophysiological measures seemed to be critical for investigating the subjective stress response in professional training.

This pattern is consistent with the psychometric measures of cognitive load obtained by NASA-TLX scores. The majority of trainees exhibited medium scores, in which frustration the domain most highly rated. Additionally, the workload scores were congruent with the performance quality determined by checklist punctuation. The participants with higher checklist scores reported less cognitive load, whereas those with lower performance scores displayed higher physical and mental processing demands. These results agree with a great deal of research demonstrating the negative correlation between task performance and the cognitive load experienced by healthcare providers in either simulation environments or clinical settings [41,47,48,49].

The relationship between pupil size variations and self-evaluated cognitive load adds more evidence to this argument. Many investigations suggest that pupil size increases according to task complexity [29,37]. In the present study, participants who self-evaluated the task as more mentally and physically demanding presented higher differences in pupil diameters during the simulation procedure. Since pupil diameter variations are closely related to parasympathetic and sympathetic impulses, increased cognitive load can trigger a rapid change in pupil diameter [37]. From this perspective, pupil diameter differences were negative and moderately associated with the temporal dimension of the NASA-TLX questionnaire. The same trend regarding the NASA-TLX temporal demand category has been observed in a previous investigation involving simulation training in a clinical environment [37].

Therefore, pupillometry seems to be a promising ocular metric for evaluating cognitive load. Previous studies have reported that pupil response precedes the physiological arousal effect even under minimal cognitive demands [50,51]. On the other hand, heart rate has also been correlated with physiological arousal when the task difficulty rises. Both pupil diameter and heart rate increase due to cognitive activity during a challenging task and can affect the performance of that task. With these considerations in mind, searching for a pattern that links pupil size variations with heart rate, cognitive load, and performance is particularly relevant. Under this assumption, our results suggest that differences in pupil diameter are associated with psychophysiological and psychometric measures. The analysis of data through a multiple regression model produced a statistically significant regression equation using pupil diameter differences as the dependent variable and pre-test blood systolic pressure, (i) heart rate, (ii) workload (physical, mental, and temporal demands) (iii), and checklist scores (iv) as predictors. The regression analysis showed a consistent pattern, resulting in a statistically significant regression equation (*p* < 0.028) and a Durbin–Watson value of 2.042, indicating the lack of autocorrelation in the sample. Consequently, our results sufficiently support the significance of the analysis to determine the validity and usefulness of the model. These findings may indicate not only the potential of pupil diameter as an indicator of cognitive activity but also the need to identify the influence of physiological arousal when performing a given task.

Although these results are encouraging, there are some limitations in the current study. One limitation is the use of convenience sampling to select participants. Future research may consider recruiting participants with different expertise and skills when performing the task. Monitoring of physiological parameters may result in a potential bias due to the perception of being evaluated. Furthermore, psychometric variables ought to be self-evaluated by participants. Further research will focus on the objective measures to determine the cognitive activity and workload experienced by participants. The experiment set did not allow monitoring of pupil diameters in short-term moments. There might have been variations in the pupillary response according to the different phases of the task, namely identification of the situation and decision-making times. Finally, cognitive activity may vary according to the contexts of decision-making tasks. Therefore, this investigation will be performed in other medical scenarios and real-life healthcare settings.

## 5. Prospects on Healthcare Training and Clinical Practice

This investigation provides more evidence regarding the potential of pupillary response to measure cognitive load in healthcare learning environments. Our analysis suggests that (i) the cognitive workload experienced by trainees in simulated clinical scenarios triggers a physiological response associated with changes in pupil diameters and variations in heart rate; (ii) workload dimensions involving mental, physical, and temporal demands are closely related to differences in pupil diameter; and (iii) clinical performance is congruent with the physiological and psychometrics measures perceived by participants.

These results are of particular interest in healthcare training for either simulated or real environments. Monitoring cognitive load appears to be crucial to maximizing clinical performance during the learning process. The utilization of objective indicators of cognitive load can help improve the simulation procedure’s effectiveness in stressful scenarios. Integrating the investigation of pupillary responses in clinical practice may lead to a better understanding of the main training problems in healthcare education. The appraisal of the trainees’ cognitive demands will contribute not only to addressing the key challenges of the cognitive process but also to improving overall performance and achieving good clinical practices.

## 6. Conclusions

The assessment of pupillary responses as indicators of cognitive load is attracting attention in healthcare education. This study provided additional evidence into the trainee’s cognitive demands involved in clinical performance through the appraisal of variations in pupil diameter. The application of a multiple regression model revealed a consistent trend between pupil diameter differences, other physiological parameters such as systolic blood pressure and heart rates, physical-mental workload, and performance scores.

These results suggest that pupil dilation is negatively associated with temporal cognitive demands and lower performance outcomes. Therefore, monitoring pupil variations during simulation exercises is of utmost importance to identify the cognitive activity of trainees. Moreover, the appraisal of clinical performance through objective metrics has implications for both learning and clinical outcomes and can ultimately lead to the improvement in the quality of healthcare. Further research is needed to validate the use of pupil diameter variations as reliable markers of physiological arousal, cognitive workload, and clinical performance.

## Figures and Tables

**Figure 1 healthcare-11-00455-f001:**
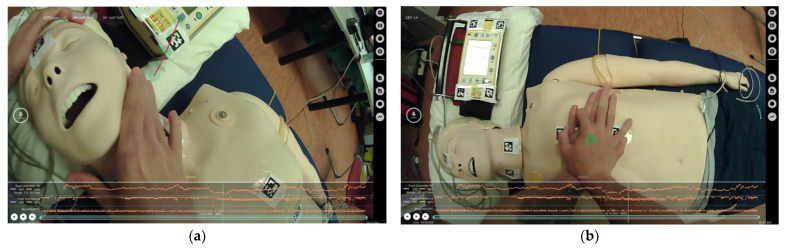
Snapshots showing pupil patterns (orange lines) and visual attention (green circles) of participants during the cardiac arrest scenario: (**a**) assessment of consciousness, breathing, and (**b**) chest compression.

**Table 1 healthcare-11-00455-t001:** Socio-demographic characteristics and checklist scores of participants.

Socio-Demographics		Socio-Demographics Values	Statistic Values X^2^/t	*p*-Value
Sex	Female	41.373 (83.7)	22.224	0.000 *
	Male	8 (16.3)		
Age		21.37 ± 5.74	69.367	0.000 *
Educational level	Baccalaureate	39 (79.6)	79.571	0.000 *
	Other Bachelor of Science Professional training Master of Science	7 (14.3) 1 (2) 2 (4.1)		
Basic life support (BLS) training	Yes	17 (34.7)	4.592	0.032 *
	No	32 (65.3)		
Type of BLS training	Online training	1 (2)	14.588	0.001 *
	On-site training	13 (26.5)		
	Mixed training	3 (6.1)		
Advanced life support (ALS) training	Yes	2 (4.1)	41.327	0.000 *
	No	47 (95.9)		
Type of ALS training	Online training	0 (0)	0.000	1.000
	On-site training	1 (2)		
	Mixed training	1 (2)		
Checklist scores		5.31 ± 2.08	17.825	0.000

* *p* < 0.05; Chi-squared Pearson; SD Standard Deviation.

**Table 2 healthcare-11-00455-t002:** Physiological and psychological variables at pre-test and post-test moments.

	Moment			
Physiological Parameter	Pre-Test	Post-Test	Statistic t-Paired	*p*-Value
Peripheral temperature	36.355 ± 0.204	36.406 ± 0.252	−1.334	0.189
Systolic blood pressure	118.90 ± 16.455	117.00 ± 14.142	1.395	0.169
Diastolic blood pressure	79.92 ± 14.527	78.43 ± 10.398	1.331	0.190
Heart rate	80.08 ± 16.4422	82.42 ± 17.133	−2.025	0.049
Oxygen saturation	97.38 ± 1.265	97.50 ± 1.011	−0.590	0.558
VAS scores	4.5 ± 2.44	4.8 ± 2.48		

**Table 3 healthcare-11-00455-t003:** Relationship of physiological and psychological variables with checklist scores.

Variables	Checklist Scores	Number of Participants	Mean ± SD	Statistic t	*p*-Value
**Pre-test measurements**					
Peripheral temperature	<5	21	36.352 ± 0.1778	−0.083	0.934
	>5	28	36.357 ± 0.2251		
Systolic blood pressure	<5	21	122.38 ± 18.096	1.257	0.217
	>5	28	116.29 ± 14.909		
Diastolic blood pressure	<5	21	84.24 ± 18.944	1.349	0.188
	>5	28	78.14 ± 9.633		
Heart rate	<5	21	83.57 ± 16.543	1.301	0.200
	>5	28	77.46 ± 15.892		
Oxygen saturation	<5	20	97.20 ± 1.399	−0.783	0.439
	>5	28	97.50 ± 1.171		
VAS score	<5	21	4.619 ± 2.1089	0.302	0.764
	>5	28	4.411 ± 2.7116		
**Post-test measurements**					
Peripheral temperature	<5	21	36.438 ± 0.1884	0.813	0.420
	>5	28	36.382 ± 0.2919		
Systolic blood pressure	<5	21	119.86 ± 13.135	1.252	0.217
	>5	28	114.86 ± 14.719		
Diastolic blood pressure	<5	21	78.71 ± 9.509	0.169	0.867
	>5	28	78.21 ± 11.186		
Heart rate	<5	20	85.65 ± 18.033	1.090	0.282
	>5	28	80.11 ± 16.396		
Oxygen saturation	<5	21	97.29 ±1.007	−1.365	0.179
	>5	28	97.68 ± 0.983		
VAS score	<5	21	5.071 ± 2.1230	0.538	0.593
	>5	28	4.696 ± 2.7532		
**Pupillary response**					
Minimum pupil diameter	<5	21	2.1576 ± 1.19611	1.157	0.257
	>5	27	1.8248 ± 0.62839		
Maximum pupil diameter	<5	21	5.6519 ± 1.39001	0.186	0.853
	>5	27	5.5789 ± 1.29099		
Pupil difference diameter	<5	21	3.4943 ± 1.39943	−0.669	0.507
	>5	27	3.7541 ± 1.24784		

**Table 4 healthcare-11-00455-t004:** Relationship of workload variables with checklist scores.

NASA-TLX Workload	Checklist Scores	Number Participants	Mean ± SD	Statistic t	*p*-Value
NASA overall	<5	21	57.5081 ± 17.63	1.337	0.188
	>5	28	50.6321 ± 17.94		
Mental demand (M)	<5	21	153.10 ± 101.99	0.330	0.743
	>5	28	144.29 ± 84.61		
Physical demand (F)	<5	21	29.29 ± 67.18	0.456	0.650
	>5	28	22.14 ± 42.17		
Temporary demand (T)	<5	21	150.71 ± 110.03	0.588	0.559
	>5	28	131.79 ± 112.98		
Performance (P)	<5	20	148.33 ± 111.46	0.390	0.698
	>5	28	136.46 ± 100.68		
Effort (E)	<5	21	151.43 ± 80.75	0.891	0.377
	>5	28	126.25 ± 108.79		
Frustration (FR)	<5	21	231.67 ± 163.36	0.839	0.406
	>5	28	195.71 ± 136.29		

**Table 5 healthcare-11-00455-t005:** Multiple linear regression analysis to model the relationship between differences in pupil diameters and physiological parameters (systolic blood pressure and heart rate pre-test), NASA-TLX workload demand (physical, mental and temporal demands), and checklist scores.

Dependent Variable: Self-Efficacy	Unstandardized Coefficients		Standardized Coefficients Beta		
	B	Standard Error		Statistic t	Significance
Constant	8.518	1.740		4.895	0.000
Systolic Blood Pressure	−0.022	0.011	−0.278	−1.948	0.058
Heart rate	−0.014	0.012	−0.171	−1.120	0.269
Mental demands	−0.004	0.002	−0.267	−1.830	0.075
Physical demands	0.002	0.004	0.075	0.534	0.596
Temporary demands	−0.005	0.002	−0.381	−2.730	0.009
Checklist scores	−0.007	0.089	−0.011	−0.079	0.938

## Data Availability

Data will be available upon reasonable request to the corresponding author.

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
