# Peer review of "Using Task-Evoked Pupillary Response to Predict Clinical Performance during a Simulation Training"

_healthcare, 2023, doi:10.3390/healthcare11040455_

Round 1
Reviewer 1 Report
Thank you for the opportunity to read this manuscript.
This is a relevant and innovative study.
In the conclusion section, the authors state: "Therefore, monitoring pupil variations during simulation exercises are of utmost importance to identifying the cognitive activity of trainees while achieving better simulation outcomes." I think this statement should be better contextualized. What is the relationship between evaluation and better results? Does this evaluation improve the simulation results as a pedagogical strategy or in terms of results for the "patient"? In what way?
As a limitation, could assessing the physiological parameters itself have increased the students' stress and affected the results obtained?
Very interesting job!
Best wishes
Author Response
Dear Reviewer,
We thank the reviewers for their constructive and valuable suggestions. Their careful reading has contributed to improving the quality of this article. According to their comments, we have comprehensively revised the manuscript to address all their requests. Following the editor’s instructions, we have included a point-by-point response to the reviewers’ comments, and the changes made have been highlighted accordingly in the revised version of the manuscript.
To help review amendments done to the manuscript, below we reproduce in full the reviewer’s comments (black color and italics font), and immediately after each comment, the reply provided in red color and portions of the revised text in the paper between quotation marks (red color and italics).
Reviewer
Thank you for the opportunity to read this manuscript.
This is a relevant and innovative study.
In the conclusion section, the authors state: "Therefore, monitoring pupil variations during simulation exercises are of utmost importance to identifying the cognitive activity of trainees while achieving better simulation outcomes." I think this statement should be better contextualized. What is the relationship between evaluation and better results? Does this evaluation improve the simulation results as a pedagogical strategy or in terms of results for the "patient"? In what way?
As a limitation, could assessing the physiological parameters itself have increased the students' stress and affected the results obtained?
Very interesting job!
Best wishes
We acknowledge the reviewer for the positive comments and valuable contributions.
Following the reviewer’s recommendation, we have modified the statement in the conclusion section:
“Therefore, monitoring pupil variations during simulation exercises is of utmost importance to identify the cognitive activity of trainees. Moreover, the appraisal of clinical performance through objective metrics has implications for both learning and clinical outcomes and can ultimately lead to the improvement of the quality of healthcare. ”
Concerning the study limitations, according to the reviewers’ recommendation, we have included the following sentence:
“Monitoring of physiological parameters may result in a potential bias due to the perception of being evaluated.”
Reviewer 2 Report
The study investigates task-evoked pupil size changes as possible additional reliable markers of mental workload and clinical performance in medical practice.
I suggest to shorten Introduction. Adequate Materials and Methods. Very clear presentation of the results. Sufficient Discussion. Critical view of the limitations of the study. Recent Reference list.
Author Response
Dear Reviewer,
We thank the reviewers for their constructive and valuable suggestions. Their careful reading has contributed to improving the quality of this article. According to their comments, we have comprehensively revised the manuscript to address all their requests. Following the editor’s instructions, we have included a point-by-point response to the reviewers’ comments, and the changes made have been highlighted accordingly in the revised version of the manuscript.
To help review amendments done to the manuscript, below we reproduce in full the reviewer’s comments (black color and italics font), and immediately after each comment, the reply provided in red color and portions of the revised text in the paper between quotation marks (red color and italics).
Reviewer
The study investigates task-evoked pupil size changes as possible additional reliable markers of mental workload and clinical performance in medical practice.
I suggest to shorten Introduction. Adequate Materials and Methods. Very clear presentation of the results. Sufficient Discussion. Critical view of the limitations of the study. Recent Reference list.
We thank the reviewer for the careful reading and positive comments. The reviewer´s suggestions are appreciated. Since the required number of words is 4000 and the manuscript count is 3733, we opt for maintaining the extension of the introduction.
Reviewer 3 Report
This is an interesting concept, using objective measures of cognitive load that will help future simulation evaluations. I have a few questions/concerns regarding the study design and the results. WIth the performance checklist, is that a validated checklist? There are various validated checklists already out there that may be of better use. Could you go back and review the videos with a validated checklist? In terms of reporting the results, you report a lot of "differences" but they are actually not statistically significant. Heart rate is the only one that is significantly increased before and after the simulation. You report a significant change in pupillary change and refer to table 2 but the values are not actually there. Also, you report that subjects with lower perfomance checklist scores (< 5) reported higher NASA scores but from table 4, the p-values are not significant. The results theoretically make sense but I don't feel you have statistically shown the relationship. While I know this was a may need a larger sample size, if possible.
Author Response
Dear Reviewer
We thank the reviewers for their constructive and valuable suggestions. Their careful reading has contributed to improving the quality of this article. According to their comments, we have comprehensively revised the manuscript to address all their requests. Following the editor’s instructions, we have included a point-by-point response to the reviewers’ comments, and the changes made have been highlighted accordingly in the revised version of the manuscript.
To help review amendments done to the manuscript, below we reproduce in full the reviewer’s comments (black color and italics font), and immediately after each comment, the reply provided in red color and portions of the revised text in the paper between quotation marks (red color and italics).
This is an interesting concept, using objective measures of cognitive load that will help future simulation evaluations. I have a few questions/concerns regarding the study design and the results. WIth the performance checklist, is that a validated checklist? There are various validated checklists already out there that may be of better use. Could you go back and review the videos with a validated checklist? In terms of reporting the results, you report a lot of "differences" but they are actually not statistically significant. Heart rate is the only one that is significantly increased before and after the simulation. You report a significant change in pupillary change and refer to table 2 but the values are not actually there. Also, you report that subjects with lower perfomance checklist scores (< 5) reported higher NASA scores but from table 4, the p-values are not significant. The results theoretically make sense but I don't feel you have statistically shown the relationship. While I know this was a may need a larger sample size, if possible.
We thank the reviewer for the careful reading and positive comments. The suggested perspectives will improve the quality of the review.
First, we appreciate the valuable suggestion regarding the selection of the checklist. To explain this aspect, we have incorporated new data about the checklist application. Briefly, we adapted the BLS Adult CPR and AED Skills Testing Checklist from the American Heart Association. Following the reviewer’s recommendation, we have included this information within cited bibliography and the supplementary material. A detailed description of the most relevant issues associated with the assessment performance has also been considered.
“The checklist version was an adaptation of the Basic Life Support (BLS) adult CPR (Cardiopulmonary Resuscitation) and AED (automated external defibrillator) skills testing checklist from the American Heart Association. The adaptation considered particular features of the proposed scenario and the previous formation of participants [41].”
[41] Cheng, A.; Magid, D.J.; Auerbach, M.; Bhanji, F.; Bigham, B.L.; Blewer, A.L.; Dainty, K.N.; Diederich, E.; Lin, Y.; Leary, M.; et al. Part 6: Resuscitation Education Science: 2020 American Heart Association Guidelines for Cardio-pulmonary Resuscitation and Emergency Cardiovascular Care. Circulation 2020, 142, S551–S579, doi:10.1161/CIR.0000000000000903.
We also acknowledge the comments concerning the reporting of results. As the reviewer accurately mentions, the comparison of physiological parameters and workload values between moments, and according to the score punctuations in the performance checklist did not reveal statistical significance differences, except for the heart rate. Therefore, we tried to explain our results by applying a multiple regression analysis to model the relationship between physiological and cognitive differences with respect to pupil variations. Contrarily to the results obtained when comparing media results, multiple linear regression analysis showed a strong relationship between pupil variations and several independent variables such as heart rate, blood pressure, performance, and various workload dimensions. Likewise, the regression analysis showed a consistent pattern resulting in a statistically significant regression equation (p < 0.028) and a Durbin-Watson value of 2.042, thus indicating the lack of autocorrelation in the sample. Consequently, we think that our results support sufficiently the significance of the analysis to determine the validity and usefulness of the model. Thus, we think that this is the study’s main contribution and according to the reviewer`s comments, we have tried to emphasize these results in the current version.
“The regression analysis showed a consistent pattern resulting in a statistically significant regression equation (p < 0.028) and a Durbin-Watson value of 2.042, thus indicating the lack of autocorrelation in the sample. Consequently, our results support sufficiently the significance of the analysis to determine the validity and usefulness of the model.”